# pH Effect on the Structure, Rheology, and Electrospinning of Maize Zein

**DOI:** 10.3390/foods12071395

**Published:** 2023-03-25

**Authors:** Yuehan Wu, Jinhui Du, Jiahan Zhang, Yanlei Li, Zhiming Gao

**Affiliations:** Glyn O. Phillips Hydrocolloid Research Centre, School of Food and Biological Engineering, Hubei University of Technology, Nanli Road, Wuhan 430068, China

**Keywords:** zein fiber film, viscosity, morphology, electrospinning

## Abstract

As a simple and convenient technology to fabricate micron-to-nanoscale fibers with controllable structure, electrostatic spinning has produced fiber films with many natural advantages, including a large specific surface area and high porosity. Maize zein, as a major storage protein in corn, showed high hydrophobicity and has been successfully applied as a promising carrier for encapsulation and controlled release in the pharmaceutical and food areas. Proteins exhibit different physical and chemical properties at different pH values, and it is worth investigating whether this change in physical and chemical properties affects the properties of electrospun fiber films. We studied the pH effects on zein solution rheology, fiber morphology, and film properties. Rotational rheometers were used to test the rheology of the solutions and establish a correlation between solution concentration and fiber morphology. The critical concentrations calculated by the cross-equation fitting model were 17.6%, 20.1%, 20.1%, 17.1%, and 19.5% (*w*/*v*) for pH 4, 5, 6, 7, and 8, respectively. The secondary structure of zein changed with the variation in solution pH. Furthermore, we analyzed the physical properties of the zein films. The contact angles of the fiber membranes prepared with different pH spinning solutions were all above 100, while zein films formed by solvent evaporation showed hydrophilic properties. The results indicated that the rheological properties of zein solutions and the surface properties of the film were affected by the pH value. This study showed that zein solutions can be stabilized to form electrospun fibers at a variety of pH levels and offered new opportunities to further enhance the encapsulation activity of zein films for bioactive materials.

## 1. Introduction

Over the past decades, electrostatic spinning, a simple and convenient technology for producing micron-to-nanoscale fibers, has produced fiber films with many natural advantages, including a large specific surface area and high porosity; thus, it has attracted a lot of attention [1]. Based on these advantages, electrostatic spinning technology has been applied to lots of fields, such as biomedical fields [2], sensors [3], and food systems [4,5]. In food systems, electrospinning of proteins and polysaccharides is usually used for carriers, food packaging materials, or sensors [6,7]. In addition, electrostatic spinning is gradually being used as a new method of bioactive substance encapsulation because of its advantages, including its ability to stretch and work continuously and its reliance on the volatilization of solvents to form fibers [8].

The Food and Drug Administration (FDA) considers maize zein, a major storage protein in corn, to be a generally recognized as safe (GRAS) and food-grade ingredient [9]. It is insoluble in anhydrous ethanol or water but is soluble in aqueous 60–90% ethanol solutions [10]. Its nontoxic and nonhazardous properties have become the subject of extensive research in the food field. Due to its high hydrophobicity, zein has been successfully applied as a promising carrier for the encapsulation and controlled release of fat-soluble compounds (e.g., gitoxin, fish oil, etc.) in the pharmaceutical and food areas [11]. Yi Wang et al. formed self-assembled microspheres by the evaporation-induced formation of solvents and investigated a range of their properties [12,13]. Many researchers have exploited the self-assembly behavior of zein to form membranes and have characterized the formed fibrous membranes using various means [14,15]. The stability and rheological properties of zein in a variety of systems have also been investigated by researchers [16,17,18].

Due to the excellent spinnability of protein-based solutions, proteins are often used as carriers of active substances [19]. Alternatively, by adding salt ions to the solution, their effect on electrospinning can be studied by the change in conductivity. The charged properties of proteins are different at different pH levels [20]. Researchers found that when working with zein at a pH level above the isoelectric point (pH = pI), proteins in the solution tend to agglomerate, but the size of individual protein molecules decreases to some extent [21,22]. Electrostatic spinning is a kind of electrohydrodynamics, and whether or not the change of the charged properties of individual polymers in solution (i.e., the change of the solution pH value) affects the spinnability of the solution and the properties of electrospun fibers remains to be explored. Different active substances, such as probiotics, antioxidants, and vitamins, require a detailed study of the spinnability and electrospun fiber properties of spinning solutions at various pH values.

Solution viscosity, surface tension, and electrical conductivity are key factors that affect the spinnability of a solution [7,23]. In electrospinning, the formation of electrospun fibers relies on the electrostatic repulsion of mutual charges and the Columbic force of the external electric field, which generate a Taylor cone of the polymer solution. Fiber formation and the morphology of fibers strongly depend on solution parameters such as concentration, molecular weight, viscosity, surface tension, and conductivity [24]. Therefore, in order to obtain homogenous nanofibers, the polymer chain interentanglement is essential to obtaining bead-free fibers [25]. The viscosity properties of the solution are the key factors in obtaining the necessary entanglement. The rheological behavior of a model of a polymer solution can be used to evaluate the entanglement state of the solution. The relationship between the entanglement state and the fiber morphology can be derived, and the morphology of the electrospun fibers can be predicted from the rheological behavior [26,27].

In this study, we constructed a model for the relationship between zein solution concentration and fiber morphology. With this model, the electrospun morphology of zein solutions at different concentrations and pH values could be quickly predicted. We also investigated the solution properties and the physicochemical properties of electrospun fibers at different pH values. The results of this study provide theoretical support for the further utilization of zein in the industry.

## 2. Materials and Methods

### 2.1. Materials

Maize zein (Z3625) was purchased from Sigma-Aldrich (St. Louis, MO, USA); hydrochloric acid (HCl); sodium hydroxide (NaOH); potassium bromide (KBr); and ethanol were purchased from Sinopharm Chemical Reagent Co., Ltd. (Shanghai, China). Deionized water was used in all experiments.

### 2.2. Preparation of Zein Solutions

Zein solutions were prepared at different concentrations (10–32.5 *w*/*v*%) by dissolving in an 80% ethanol aqueous solution under constant stirring using a magnetic stirrer until homogeneous at room temperature (25 °C). Then, the pH of the solution was adjusted to 4, 5, 6, 7, and 8 using 0.01 mol/L HCl and NaOH, and it was stirred magnetically by a magnetic stirrer for 2 h.

### 2.3. Rheology Measurements

Rheological measurements were performed using a HAAKE Rheostress 6000 rheometer (Thermo Scientific, Karlsruhe, Germany) with a 60-mm-diameter double cone/plate sensor (C60/1°Ti L). The flow behavior (shear stress, shear rate) was recorded at 25 ± 0.5 °C in controlled shear mode using steady-state conditions between 0.01 and 1000 s^−1^. Haake Rheowin^®^ Job Manager software was used to perform the curve fitting.

### 2.4. Electrospinning Process

The electrospinning device was placed horizontally, using a 5 mL syringe connected with a 21G needle with an inner diameter of 0.5 mm, and the injection rate was 6 mL/h. The voltage of the high-voltage DC power supply was 15 kV, and the distance between the needle and the tin collector was 15 cm. Throughout the experiment, the temperature and relative humidity were controlled at 25 ± 2 °C and 50 ± 5%, respectively, and the air bubbles inside the needle were discharged before the experiment.

### 2.5. Morphology of Electrospun Fibers

The morphology of electrospun fibers was collected using a scanning electron microscope (SEM; JEOL, JSM-6390LV, Akishima-shi, Japan) after they were sputtered with a gold–palladium mixture in vacuum. We randomly selected 100 individual fibers and analyzed their diameters using ImageJ software.

### 2.6. Characterization

The conductivity of zein solutions was measured using a benchtop conductivity tester at room temperature and expressed in μS/cm. Fourier-transform infrared spectroscopies (FT-IR) of the samples were recorded at ambient temperature on a FT-IR spectrometer (ThermoFisher Scientific, Nicolet iS10, Waltham, MA, USA) in transmittance mode in the range of 4000–400 cm^−1^; the wavenumber resolution and scan number for all samples were 4 cm^−1^ and 64 scans, respectively. The electrospun films were cut into small pieces and dried overnight in an oven at 40 °C, then mixed and ground into powders with potassium bromide; the mixture was pressed to form a sample disk for the tests. The secondary structure changes of zein powder and zein film obtained by evaporation and electrospinning were analyzed using Omnic software (version 8.0, Thermo Nicolet Inc., Waltham, MA, USA) and Peakfit software (version 4.12, SPSS Inc., Chicago, IL, USA) to identify the amide I band (1600–1700 cm^−1^) in each spectrum.

The X-ray diffraction (XRD) spectra were obtained using an X-ray diffractometer (Rigaku D/Max-3B, Tokyo, Japan) with a Cu-Kα emission source (λ = 1.54056 Å, 40 kV, 100 mA), 2θ scan range of 5° to 40°, a scan rate of 5°/min, and a test temperature of 25 °C. 

The water contact angle of the films was tested using a contact angle meter (Dataphysics OCA 15EC, Filderstadt, Germany). The electrospun film was cut to a size of 5 cm × 5 cm, and 2 µL of distilled water was dropped on the surface using a microsyringe for the water contact angle test. The spun solutions with different pH values were poured into Petri dishes and dried at 30 °C for 48 h; the formed films were used for the water contact angle test. 

The morphology and surface roughness of the zein fiber film were tested using an atomic force microscope (AFM). The AFM testing was selected in scanasyst-air mode, using a silicon nitride (Si_3_N_4_) probe (Bruker, Karlsruhe, Germany) attached to a cantilever with a vibration frequency of 70 kHz and a spring constant of 0.4 N/m. The scanning frequency was 1 Hz. We covered the microscope with sound insulation to reduce the interference of vibration during the test. After adhering the mica sheet to the substrate and starting electrospinning, the sample was placed on a stable receiving tin collector for 15 s and then placed in a sealed box for testing. The NanoScope Analysis 1.9 software was used to analyze all the AFM images.

### 2.7. Statistical Analysis

The results were performed for at least three samples (n ≥ 3) in order to confirm the reproducibility of the experimental results and were expressed as mean ± standard deviation. The results were evaluated using one-way ANOVA. A value of *p* < 0.05 was considered statistically significant.

## 3. Results and Discussion

### 3.1. Rheological Properties

In electrospinning, solution rheology, especially solution viscosity, which is associated with the polymer molecular weight and polymer concentration, is a significant parameter affecting the process behavior and the morphology and mechanical properties of the obtained fiber films [26]. The rheological properties of a polymer solution depend on the interaction between the polymer and the solvent. In Figure 1, it is obvious that the increment of the solution viscosity increased with the increase in zein concentration, as was observed in all the samples. As zein concentration increases, the molecular chains begin to entangle and overlap, which reduces chain re-ordering behavior; therefore, the mobility of zein molecular chains is gradually reduced and the solution viscosity increases. Zein is a kind of protein, and besides the molecular weight and concentration, pH is also an important factor for the rheology of a protein solution. In order to investigate the influence of spinning solution pH on the properties of the formed fiber films, the rheological properties of zein solutions at different concentrations and five different pH levels were measured. As shown in Figure 1, the solutions showed shear thinning characteristics at low shear rates, with the viscosity approaching zero shear rate increases, which might be due to the presence of aggregates. When the shear rate gradually increased in the range of 1–1000 s^−1^, all of the spinning solution ratios showed Newtonian fluid, and the viscosity did not change with shear rate [28]. This result indicated that the intermolecular forces between the molecules of zein solutions at different pH and concentrations were weak, and at lower shear, the intermolecular forces were all broken, thus revealing the properties of a Newtonian fluid at higher shear rates. Electrospun nanofibers are normally organized from polymeric solutions; a stable electrospinning jet that needs proper polymer chain entanglements (or critical entanglement concentration, C_e_) is necessary for the formation of uniform nanofibers [26,29]. 

In order to investigate the critical entanglement concentration of the zein solution with different pH values and further study the relationship between fiber morphology and pH of the zein solution, we fitted the rheological results for different concentrations at different pHs using the cross-equation fitting model (Equation (1)) in the data processing software to obtain the zero shear viscosity (*η*_0_) [30]:(1)$$\frac{\eta -\eta_{\infty }}{\eta_{0}-\eta_{\infty }}=\frac{1}{1+\alpha {\dot{\gamma }}^{m}}$$
where *η_∞_* is the apparent viscosity at an infinite rate of shear, assumed to be 0 by the software; α and $\dot{\gamma }$ are two independent constants; and *m* is referred to as the rate constant. Then the specific viscosity was calculated according to the following equation (Equation (2)):
*η_sp_* = (*η*_0_ − *η_s_*)*/η_s_*,(2)
where *η_s_* is the viscosity of the solvent, which is the viscosity of an 80% (*v*/*v*) ethanol aqueous solution at different pHs. In Figure 2, the relationship between specific viscosity and concentration at all pH levels was divided into three intervals, which were the semidilute unentangled region, the semidilute entangled region, and the concentrated region [26]. In the semidilute unentangled region, the polymer chains in the solution existed in an independent manner and were not entangled. The overlap concentration (C_e_) was the dividing point between the semidilute unentangled region and the semidilute entangled region and was the lowest concentration at which electrospun fibers could produce continuous fibers. We calculated the intersection of the different regions to obtain the critical concentrations of 17.6%, 20.1%, 20.1%, 17.1%, and 19.5% (*w*/*v*) for pH 4 to 8. In the semidilute entangled region, an initial overlap between polymer solvent molecules occurred. As the concentration increased, entanglement began between the polymer chains wrapped by the solvent, and the inflection point of the specific viscosity change marked the beginning of the concentrated region (C^**^). The starting concentrations in the concentrated region were relatively similar at different pH levels, all around 30%. However, pH 6 was closer to the isoelectric point (pI) of the zein solution, and the aggregated precipitation of the solution led to a larger value of the obtained viscosity. As a result, the starting point (C^**^) of its concentrated region occurred earlier.

At low concentrations, the concentration dependence of zein protein solutions with different pH was low with indices of *η_sp_*~c^0.68^, *η_sp_*~c^0.68^, *η_sp_*~c^0.66^, *η_sp_*~c^0.66^, and *η_sp_*~^0.58^, respectively, which was similar to the indices (*η_sp_*~c^0.5^) of classical polyelectrolytes in the semidilute unentangled state [31]. When the concentration increased, the degree of overlap and entanglement between polymer molecules increased, and the concentration dependence was enhanced.

### 3.2. Conductivity of Solution

Electrical conductivity is an important factor that affects the formation and morphology of the electrospun fibers. It determines charge mobility, which influences electrostatic repulsion force and thus the morphology of electrospun fibers [24]. We investigated the variation in solution conductivity of proteins at different pHs and the effect of this variation on fiber morphology. As shown in Figure 3, all samples had high conductivity at 30% concentration, with a small decrease when the pH of the solution is six. In the apparent viscosity of the five samples, we observed some increase when the pH of the solution was six. These changes could be attributed to the isoelectric point (pI = 6.2) of zein proteins. When the protein was at the isoelectric point, the net charge of the protein molecules was zero, the conductivity of the solution was low, and it was easy for aggregation between the protein molecules to occur.

### 3.3. Morphology of Zein

The morphology of electrospun fiber film is correlated with the entangled state of the polymer molecules in the spinning solution. In order to investigate the relationship between the morphology of zein fiber film and zein solution, we chose three concentrations: 12.5%, 15%, and 30% (*w*/*v*) based on the plot of specific viscosity versus concentration shown in Figure 2, which was in the semidilute unentangled region, the semidilute entangled region, and the concentrated region, respectively. In Figure 4, the SEM images showed that as the concentration increased, the electrospun fibers changed from granular to beaded fibers and finally to bead-free fibers. The entanglement state of the solution had a key effect on the morphology of the fiber. The three regions represented the different solution states and the three morphologies of the fiber that corresponded to the three regions shown in the figure. As mentioned before, the polymers in the solution existed in an independent manner and were not entangled when the concentration was in the semidilute unentangled region. As shown in the first line of Figure 4, all the samples appeared granular at low viscosity; these results agreed with the results obtained in the specific viscosity analysis. Electrospray behavior was observed because the molecular chains were unable to entangle at this concentration. As the concentration increased, entanglement began between the polymer chains wrapped by the solvent, as shown in the second line of Figure 4, and beaded fibers were observed in all the samples. Due to the molecular chain entanglement, jet fracture caused by the electrostatic field during the electrospinning process was prevented; hence, the electrospray behavior gradually changed to electrospinning behavior. As the concentration further increases, in the third line of Figure 4, bead-free fibers were observed at all pH values when the concentration of zein was 30% (*w*/*v*). In this condition, the molecular chain entanglement and overlap in the spinning solution were high enough for the specific processing condition to form uniform bead-free fibers. It was evident from the SEM plots that the morphology of maize alcoholic proteins changed with concentration in agreement with the predicted values of rheology. In contrast, the change in pH did not significantly affect the morphology of the fibers. Although the onset of the concentration region was slightly advanced at pH 6, in practice, the concentration at the inflection point did not appear as perfect bead-free fibers. The increase in viscosity at this point was not caused by polymer molecular polymerization but rather by the tendency of zein protein particles to polymerize and precipitate near the isoelectric point. This effect led to an increase in viscosity, which in turn led to a change in the fitting results. We tested the fiber diameter of the bead-free fibers produced at 30% concentration using ImageJ software. As shown in Figure 5, it is obvious that the average diameter of the fibers was a maximum of 0.476 μm at pH 8 and a minimum of 0.321 μm at pH 5. At 30% (*w*/*v*) concentration, bead-free fibers were produced at all pH levels. Therefore, we performed the subsequent tests using fiber mats made at a 30% (*w*/*v*) concentration.

### 3.4. XRD 

During the electrospinning process, the molecular chains of zein could be stretched and solidified rapidly by the electric field stress, thereby promoting the formation of an amorphous structure and impeding the crystallization. According to our results, as mentioned before, we found that pH could affect the conductivity of the spinning solution and further affect the electrostatic repulsion force during the electrospinning process. Hence, pH might also affect the molecule conformation of the zein fiber film. XRD is an important tool used to study the microstructural changes of substances and was used to investigate the effect of pH on the conformation of molecules in electrospun fibers in detail. As shown in Figure 6, zein electrospinning mats with different pH values presented an amorphous structure, and the two larger diffraction peaks at 9.1° and 19.8° were consistent with those reported in the literature [32,33]. By applying Bragg’s law, the corresponding spacings of the two diffraction peaks were 10.1 Å and 4.6 Å. According to the literature, these two spacings corresponded to the pitch and stacking distance of the α-helix [34]. When the pH was above the isoelectric point, the pitch and stacking distance of the α-helix of the fiber felt were greatly reduced, as shown in Figure 6. This result indicated that pH above the isoelectric point affected the structure of the protein.

### 3.5. Secondary Structure Analysis

Fourier transform infrared spectrometry (FT-IR) is an effective means to determine the secondary structure of proteins. It could detect the conformational changes of zein induced by the variation in pH values. The secondary structural elements of polypeptide chains such as the α-helix, β-sheet, β-turn, and random coil could be characterized by correlating the frequencies of amide modes to specific types of hydrogen-bonding patterns [35]. The amide I region (1600–1700 cm^−1^) contains abundant secondary structure information, such as the α-helix, β-sheet, β-turn, and random coil structure [18,36]. Therefore, the amide I region is commonly used to analyze the secondary structure of proteins. The secondary structure distribution of the amide I region was as follows: β-sheet (1620–1640 cm^−1^, 1675–1700 cm^−1^), random coil (1640–1648 cm^−1^), α-helix (1650–1660 cm^−1^), and β-turn (1660–1675 cm^−1^) [18]. We processed and calculated the FT-IR spectral data for pure zein powder and zein films formed by solvent evaporation and electrospinning to obtain the protein secondary structure distribution ratios. As shown in Table 1, the secondary structures of pure zein powder and electrospun fiber film were significantly different. Compared with five electrospun zein fiber films, when pH was 6, the secondary structure changed most obviously, the relative content of β-sheet increased significantly, and the relative content of α-helix decreased significantly. These results showed that the secondary structure of the protein near the isoelectric point changed significantly; this was attributed to the α-helix and some random coil structures having been changed, which increased the relative content of the β-sheet. 

### 3.6. AFM Analysis

According to the results of previous studies, the different ratios of ethanol and water in the solvent affected the evaporation rate of the solvent. When the ratio of ethanol in the solvent gradually increased, the three-dimensional image of electrospun fibers changed from cylindrical to elliptical and then to a strip. According to the AFM image observation, the electrospinning fibers under different pH values were elliptical because of the consistent ratio of ethanol and water, which was consistent with the previous research results (Figure 7). However, the particle sizes of proteins in the solution under different pH values were different [21,37], which resulted in certain differences in the morphology of the different fibers. When the pH value was 4 or 5, because of the large protein particles in the solution, the formed fibers were rough. When the pH value was above the isoelectric point, the particle size decreased, and the fibers became relatively smooth.

### 3.7. Water Contact Angle

Surface hydrophobicity is an important physical property of fiber films. The characteristics of the sample and the roughness of the surface were the key factors affecting surface hydrophobicity [38]. The surface hydrophobicity of a fiber film is usually measured by the water contact angle. During the electrospinning process, we found that the electrospun fiber films had good water absorption at all pH values. Therefore, we measured the contact angle that formed after water droplets were in contact with the fiber membrane surface for 0.5 s. As shown in Figure 8a, it was clear that the instantaneous contact angles of the fiber membranes prepared with different pH spinning solutions were all above 100°. With an increase in solution pH, the contact angle decreased to some extent, which was consistent with the results of Wang et al. [39]. The roughness of the fiber surface was analyzed using AFM, as shown in Table 2. The surface roughness of the fibers formed by electrospinning decreased at pH 6 and increased slightly at pH 8, which was consistent with the trend of the contact angle of the fiber film. In addition, in the range of pH values above the isoelectric point, zein tended to aggregate and precipitate, and the particle size became smaller, which led to a certain degree of reduction [20,21]. Meanwhile, as an amphiphilic corn protein, zein has an almost equal distribution of hydrophobic and hydrophilic amino acids. Thus, zein can exhibit different hydrophilic and hydrophobic properties under different conditions [40], and pH affects its hydrophilic and hydrophobic properties [41]. The water contact angles of the zein films formed by solvent evaporation were also tested, as shown in Figure 8b. For solutions with different pH values, the films formed by solvent evaporation all exhibited hydrophilic surface properties, and the hydrophilicity of the films was enhanced when the pH value was higher than the isoelectric point. The surface hydrophilic properties of the zein films formed by solvent evaporation followed similar trends to those of the zein fiber films. Therefore, the combined effect of the solution pH and fiber roughness difference led to changes in the contact angle of the zein fiber films.

## 4. Conclusions

In conclusion, we investigated the conductivity and rheological properties of zein solutions as well as the physicochemical properties of electrospun zein fiber films. First, we obtained a model of the correlation between the morphology of the zein film and the concentration of the solution using the cross-equation fitting model. Second, when the solvent pH value was equal to the protein isoelectric point, the solution viscosity and conductivity changed significantly because of the interaction between protein molecules and the change in charge properties. Finally, the surface properties of zein films were influenced by the surface roughness of the films and the distribution of hydrophobic groups at different pH values. In short, a zein solution with a high concentration had good spinnability, and a change in the solution’s pH value had less effect on its spinnability. The change in the pH value, however, had a significant influence on the chemical and physical properties of the fiber film.

## Figures and Tables

**Figure 1 foods-12-01395-f001:**
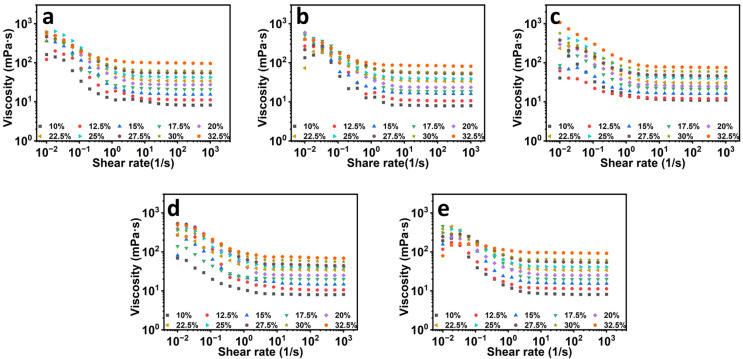
The rheological properties of zein solutions at different concentrations at pH values of 4 (**a**), 5 (**b**), 6 (**c**), 7 (**d**), and 8 (**e**).

**Figure 2 foods-12-01395-f002:**
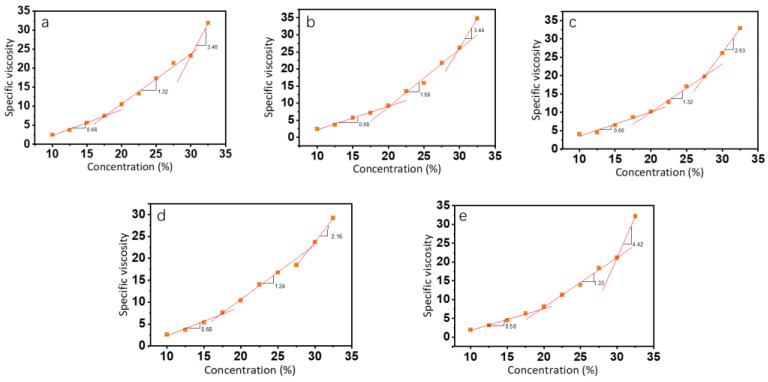
The relationship between specific viscosity and concentration of zein at pH values of 4 (**a**), 5 (**b**), 6 (**c**), 7 (**d**), and 8 (**e**).

**Figure 3 foods-12-01395-f003:**
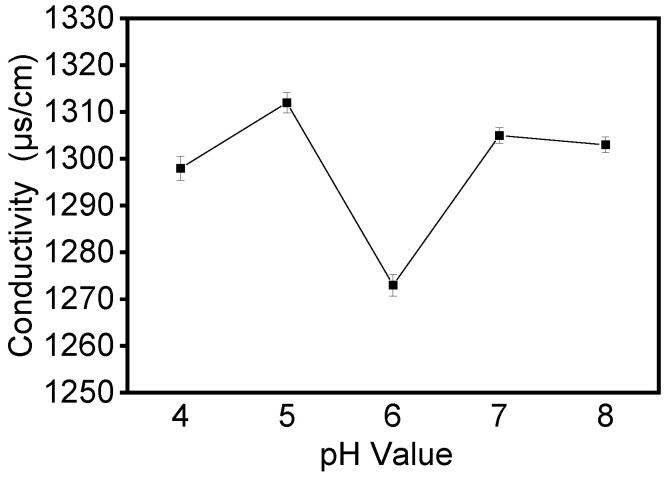
The conductivity of the zein film obtained by zein solution at different pH values.

**Figure 4 foods-12-01395-f004:**
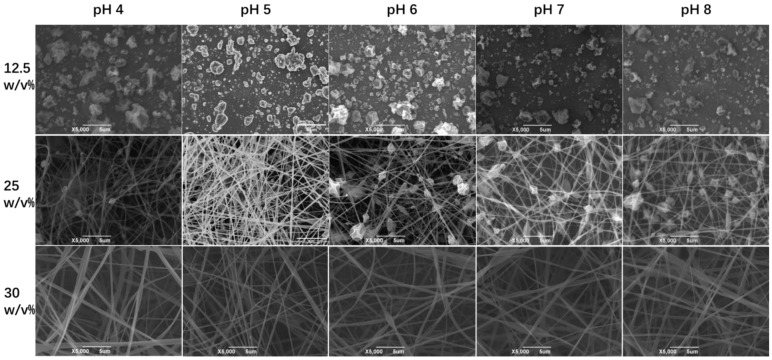
The SEM images of the surface morphologies of the zein films obtained by zein solutions with different concentrations at different values.

**Figure 5 foods-12-01395-f005:**
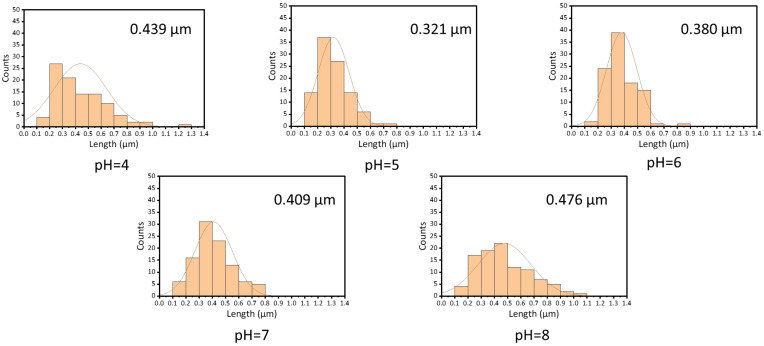
The diameter of the zein film is obtained by using zein solutions with different pH values.

**Figure 6 foods-12-01395-f006:**
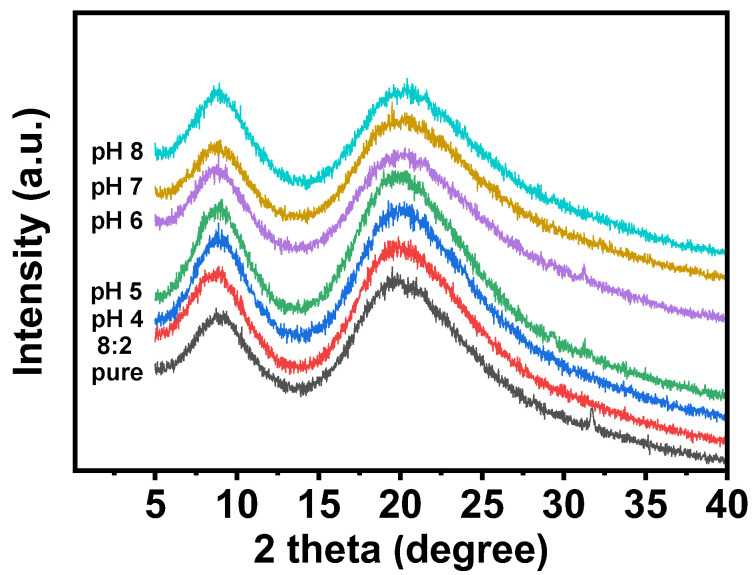
The XRD spectra of zein powder and zein film are obtained by using zein solutions with different pH values.

**Figure 7 foods-12-01395-f007:**
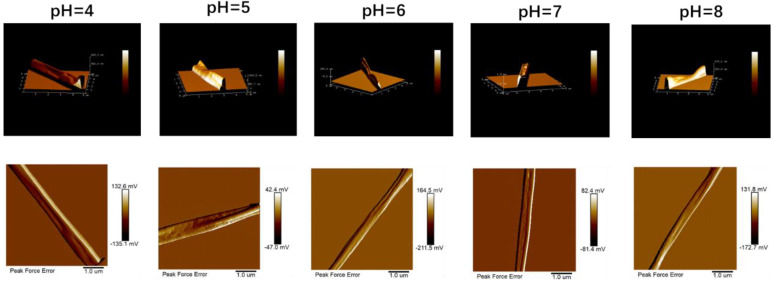
The AFM image of electrospun fibers is obtained by zein solutions with different pH values.

**Figure 8 foods-12-01395-f008:**
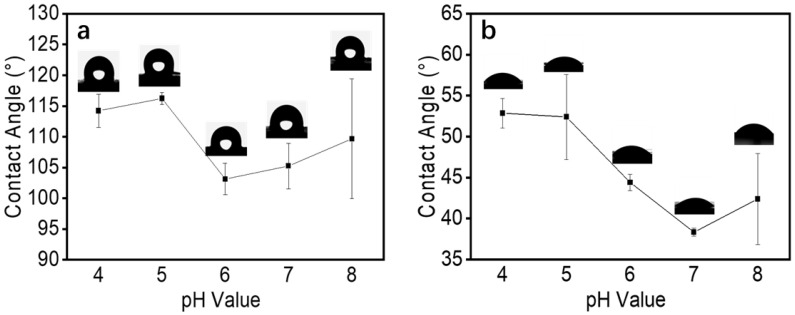
(**a**) The contact angle of electrospun zein film obtained by zein solutions with different pH values. (**b**) The contact angle of the zein film formed by solvent evaporation.

**Table 1 foods-12-01395-t001:** Calculated secondary structures of pure zein and zein films (spectra used in Appendix A).

Sample	α-Helix (%)	β-Sheet (%)	β-Turn (%)	Random Coil (%)
Pure zein	17.63 ± 0.30 ^g^	42.97 ± 0.72 ^b^	21.00 ± 0.30 ^a^	18.40 ± 0.15 ^b^
8:2 film	19.45 ± 0.11 ^e^	45.15 ± 0.16 ^a^	20.95 ± 0.15 ^a^	14.45 ± 0.11 ^d^
pH = 4	26.74 ± 0.07 ^b^	36.72 ± 0.08 ^c^	18.20 ± 0.09 ^cd^	18.27 ± 0.10 ^bc^
pH = 5	26.16 ± 0.07 ^c^	35.87 ± 0.10 ^d^	18.36 ± 0.17 ^c^	19.62 ± 0.20 ^a^
pH = 6	18.58 ± 0.08 ^f^	45.47 ± 0.10 ^a^	17.90 ± 0.12 ^de^	18.00 ± 0.10 ^c^
pH = 7	28.10 ± 0.20 ^a^	34.64 ± 0.07 ^e^	17.82 ± 0.10 ^e^	19.45 ± 0.24 ^a^
pH = 8	23.38 ± 0.09 ^d^	36.95 ± 0.12 ^c^	20.03 ± 0.21 ^b^	19.64 ± 0.17 ^a^

Different letters in each row indicated significant differences (*p* < 0.05).

**Table 2 foods-12-01395-t002:** Surface roughness of electrospun fibers obtained by zein solutions with different pH valuess.

pH	4	5	6	7	8
Rq	140 ± 5.29	128.33 ± 6.66	78.33 ± 8.25	76.53 ± 8.42	105.1 ± 17.94
Ra	125.67 ± 5.51	111.33 ± 7.23	67.4 ± 9.71	70.73 ± 8.76	97.13 ± 17.06
Image Rq	108	107	40.5	40.9	63.7
Image Ra	75	69.3	19	20.7	34.4

## Data Availability

The data in this research are available upon request from the corresponding author.

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
