# Peer review of "pH Effect on the Structure, Rheology, and Electrospinning of Maize Zein"

_foods, 2023, doi:10.3390/foods12071395_

Round 1
Reviewer 1 Report
the investigations of the rheological properties and conductivity of the zein solutions, as well as the physicochemical properties of electrospun zein films was conducted.
It is original and relevant topic, as well explained.
According to the manuscript and cited references the theme can provide innovative information.
The references are relevant and the figures presented in the text are well presented and provide information on important points of the project.
Keywords: Zein; Rheology; Electrospinning. It is recomended keywords different from title.
lines 26, 34 not coma
line 174 figure 4. (not correct) Figure 4, (correct)
line 193 (Figure S1) could be in the text
line 203 detail.
line 204 Figure 5,
line 265 s (Figure S3) could be in the text
Author Response
Thank the reviewer for reviewing our manuscript very much. We express our appreciation for their critical comments and kind suggestions to improve this manuscript. The manuscript has now been revised according to other comments.
1. Keywords: Zein; Rheology; Electrospinning. It is recomended keywords different from title.
Authors reply: Thanks for your kind suggestion, we have revised the keyword.
2. lines 26, 34 not coma
Authors reply: We converted the commas to periods.
3. line 174 figure 4. (not correct) Figure 4, (correct)
Authors reply: We have revised the format.
4. line 193 (Figure S1) could be in the text
Authors reply: Thanks for your kind suggestion, we have adjusted Figure S1 to a proper position in the text.
5. line 203 detail.
Authors reply: We have revised the punctuation.
6. line 204 Figure 5,
Authors reply: We have revised the format.
7. line 265 s (Figure S3) could be in the text
Authors reply: Thanks for your kind suggestion, we have also adjusted Figure S3 to a proper position in the text.
Reviewer 2 Report
The manuscript was well written and presented. The contents of the work are similar to what has been done on electrospinning. The discussion is good, presenting the usual explanations for this line of work. The work presents good data and refers to a case study, not presenting much novelty.
Further comments:
1. Please increase the size of Figure 1. It is too small. It is impossible to read the legends.
2. Results. Line 125. Which is the critical concentration? Please inform the value.
3. Results. Figure 5. Please increase the width of the lines of the legend. It is difficult to visualize the colors of each one.
Author Response
Thank the reviewer for reviewing our manuscript very much. We express our appreciation for their critical comments and kind suggestions to improve this manuscript. The manuscript has now been revised according to other comments.
The manuscript was well written and presented. The contents of the work are similar to what has been done on electrospinning. The discussion is good, presenting the usual explanations for this line of work. The work presents good data and refers to a case study, not presenting much novelty.
Authors reply: Thanks for your comments. The existing studies mainly focused on the formation, morphology, and structure of nanofibers by methods of mixing other compounds, emulsions, and coaxial electrospinning or physical, chemical, and enzymatic alteration of zein nanofibers. Contrarily, few studies explored the simplest two-compound electrospinning system (solvent and solute compound) on zein. Therefore, we considered it worth noticing the relationship between the spinning zein solution rheology property and the properties of formed zein fiber films, and investigating how the pH influences the rheology of zein solution and further influence the properties of the zein fiber.
Further comments:
1. Please increase the size of Figure 1. It is too small. It is impossible to read the legends.
Authors reply: Thanks for your kind suggestion, we have revised Figure 1 and adjusted all the characters to the proper sizes.
2. Results. Line 125. Which is the critical concentration? Please inform the value.
Authors reply: Electrospun nanofibers are normally organized from polymeric solutions, a stable electrospinning jet that needs proper polymer chain entanglements (or critical entanglement concentration, Ce), and uniform fibers need a stable electrospinning jet. The specific value of the zein solution with different pH was informed in the next paragraph, which was 17.6%, 20.1%, 20.1%, 17.1%, and 19.5% (w/v) for pH 4, 5, 6, 7, 8, respectively.
3. Results. Figure 5. Please increase the width of the lines of the legend. It is difficult to visualize the colors of each one.
Authors reply: Thanks for your kind suggestion, we have revised Figure 5. We moved the legend labels next to the lines and increased the font size.
Reviewer 3 Report
The manuscript aims to highlight the “pH Effect on the Structure, Rheology, and Electrospinning of Zein”. The data of this paper were analyzed and discussed properly.
I have the following comments for the authors to consider:
1. Give some values of results in the abstract section.
2. Line 66, Page 2: “Zein was from purchased”. Correct it as “Zein was purchased from”
3. Discuss the application part of the zein solution on the basis of rheological properties in section 3.1.
Author Response
Thank the reviewer for reviewing our manuscript very much. We express our appreciation for their critical comments and kind suggestions to improve this manuscript. The manuscript has now been revised according to other comments.
The manuscript aims to highlight the “pH Effect on the Structure, Rheology, and Electrospinning of Zein”. The data of this paper were analyzed and discussed properly.
Authors reply: Thanks for your comments.
Give some values of results in the abstract section.
Authors reply: Thanks for the kind suggestion. We have revised the abstract section and put some key data in it.
Line 66, Page 2: “Zein was from purchased”. Correct it as “Zein was purchased from”
Authors reply: Thanks for the kind suggestion, we have revised it.
Discuss the application part of the zein solution on the basis of rheological properties in section 3.1.
Authors reply: Thanks for your kind suggestion. We have revised the ‘3.1. Rheological properties’ part, and added more information about the application of zein related to its rheological properties
Reviewer 4 Report
Dear Authors,
Comments on your manuscript are as follows.
L.2, 66
Please put scientific name.
L.51, 52, 70, 246, 260
Reference(s) is missing. Please support your statement with relevant references.
L.116-117
What is the basis for the relationship between solution being coalesced and precipitated and shear thinning characteristics.
L.120-122, 123
Need reference(s). Please be specific.
L.225 “significantly”
Please report spread of data (i.e., Stdev., SE) and significant level of value in Table 1.
L.250
Where is the result?
L.263
Please elaborate in the Material & Method section.
Author Response
Thank the reviewer for reviewing our manuscript very much. We express our appreciation for your critical comments and kind suggestions to improve this manuscript. The manuscript has now been revised according to other comments.
L.2, 66 Please put scientific name.
Authors reply: Thanks for your kind suggestion, zein is the scientific name of a class of prolamine protein found in corn.
L.51, 52, 70, 246, 260 Reference(s) is missing. Please support your statement with relevant references.
Authors reply: Thanks for your kind suggestion, we have added the relevant references to support these sentences.
L.116-117 What is the basis for the relationship between solution being coalesced and precipitated and shear thinning characteristics.
Authors reply: Thanks for your comment, we have already revised this part. The pH could affect the aggregation of the zein solution, thus we supposed that the viscosity approaching to zero-shear rate increases which might be due to the presence of aggregates. Many studies found the same result about the shear thinning characteristics at low shear rates, but they don’t have a plausible explanation with direct evidence.
L.120-122, 123 Need reference(s). Please be specific.
Authors reply: Thanks for your kind suggestion, we have added the relevant references to support these details.
L.225 “significantly” Please report spread of data (i.e., Stdev., SE) and significant level of value in Table 1.
Authors reply: Thanks for your kind suggestion, we have revised Table 1 with more information.
L.250 Where is the result?
Authors reply: The good water absorption ability was found during the preparation process. Based on this phenomenon, we tested the water contact angle of the fiber film, and the result is shown in figure 8a.
L.263 Please elaborate in the Material & Method section.
Authors reply: We have revised this part and moved the operation part to 2.6 characterization part.
Round 2
Reviewer 4 Report
Authors did not include Zein's scientific name as suggested (comment 1).
Author Response
Thank the reviewer for reviewing our manuscript very much. We express our appreciation for your critical comment to improve this manuscript. The manuscript has now been revised.
Comments: Authors did not include Zein's scientific name as suggested (comment 1).
Authors reply: Thanks for your comment. Zein used in this manuscript was purchased from Sigma-Aldrich, it was extracted from Zea mays (Maize). The product name is Zein, and the product number is Z3625. We found relevant information of zein (Z3625) on the UniProt website, and the recommended name is ‘Protein FLOURY 2’ or ‘22 kDa alpha-zein 16’ which are difficult to identify. Furthermore, most people use ‘zein’ or ‘maize zein’ as its name in their research articles, ‘maize zein’ is more specific because it provides information on the biological source. Hence, we changed ‘zein’ to ‘maize zein’ in the title and main body of this manuscript.